# Helicase promotes replication re-initiation from an RNA transcript

Bo Sun [1,2,4], Anupam Singh[3], Shemaila Sultana[3], James T. Inman[1,2], Smita S. Patel[3] & Michelle D. Wang [1,2]

To ensure accurate DNA replication, a replisome must effectively overcome numerous obstacles on its DNA substrate. After encountering an obstacle, a progressing replisome often aborts DNA synthesis but continues to unwind. However, little is known about how DNA synthesis is resumed downstream of an obstacle. Here, we examine the consequences of a non-replicating replisome collision with a co-directional RNA polymerase (RNAP). Using single-molecule and ensemble methods, we find that T7 helicase interacts strongly with a non-replicating T7 DNA polymerase (DNAP) at a replication fork. As the helicase advances, the associated DNAP also moves forward. The presence of the DNAP increases both helicase's processivity and unwinding rate. We show that such a DNAP, together with its helicase, is indeed able to actively disrupt a stalled transcription elongation complex, and then initiates replication using the RNA transcript as a primer. These observations exhibit T7 helicase's novel role in replication re-initiation.

[1] Howard Hughes Medical Institute, Cornell University, Ithaca, NY 14853, USA. [2] Physics Department & LASSP, Cornell University, Ithaca, NY 14853, USA. [3] Department of Biochemistry & Molecular Biology, Robert Wood Johnson Medical School, Rutgers University, Piscataway, NJ 08854, USA. [4] Present address: School of Life Science and Technology, ShanghaiTech University, Shanghai 201210, China. Correspondence and requests for materials should be addressed to B.S. (email: sunbo@shanghaitech.edu.cn) or to M.D.W. (email: mwang@physics.cornell.edu)

Replication fork arrest or collapse occurs when the replisome encounters obstacles such as DNA damage, DNA-bound complexes, or stable DNA secondary structures[1–4]. In addition, as replication and transcription proceed simultaneously on the same template DNA, the two must inevitably collide. In fact, many lines of evidence in vitro and in vivo support the occurrence of both co-directional and head-on collisions[5–8]. Several mechanisms have evolved to avoid replication failure from fork barriers and to restart the arrested replication forks in ways that are independent of the replication origins. The existence of these multiple pathways highlights the importance of replication fork recovery[7, 9–11]. However, the various mechanisms of fork restart are still unclear. Upon encountering a lesion on the leading strand and in the absence of lesion bypass, a replisome is often found to terminate replication at the lesion, but continues DNA unwinding[12–15], and this necessitates restarting DNA synthesis downstream from the lesion. A prerequisite for a replisome to resume replication after the lesion is the acquisition of a primer, which allows DNA polymerase (DNAP) to re-initiate DNA synthesis. Although re-priming by primase has been shown to rescue replication[16], this occurs at relatively low efficiency in overcoming leading-strand lesions and is not adopted by many organisms. There are likely additional replisome recovery pathways to facilitate primer acquisition for replication re-initiation. Replication forks can pick up primers from stable R-loops or stalled transcriptional elongation complexes[17, 18]. However, the mechanism of replication fork restart by this pathway is not understood.

Here, we investigate this replication re-initiation pathway using the bacteriophage T7 replisome. The bacteriophage T7 replisome is a simple model system which provides a powerful in vitro system to decipher detailed mechanisms of DNA replication[19–23]. It consists of T7 DNAP (gp5 protein), T7 helicase-primase (gp4), processivity factor Escherichia coli thioredoxin (trx), and the single strand binding protein (gene 2.5 protein). For simplicity, here the gp5/trx complex has been referred to as DNAP. Bacteriophage T7 itself lacks translesion polymerases to perform translesion synthesis directly on lesion sites[24]. However, we recently demonstrated that T7 DNAP, working in conjunction with helicase through specific helicase–DNAP interactions, is able to replicate through a leading-strand cyclobutane pyrimidine dimer (CPD) lesion[15] and this has been observed in other systems[25]. Such a direct lesion bypass event occurred in only about 28% of the T7 replisomes, while the remaining population continued helicase unwinding without DNA synthesis beyond the lesion. This suggests the possible existence of other mechanisms for replication re-initiation downstream of the damage.

In this report, we address the questions of whether and how a non-replicating T7 DNAP in the presence of helicase could use a nascent RNA transcript from an RNAP polymerase (RNAP) as a primer to re-initiate DNA replication. Non-replicating T7 DNAP is herein defined as DNAP in a state that is not replicating either due to lack of a complete set of dNTPs or lack of an extendable primer. We find that a non-replicating T7 DNAP interacts strongly with a T7 helicase at a fork, and this interaction significantly reduces helicase slippage frequency, leading to a faster and more processive unwinding. Furthermore, T7 helicase in association with the non-replicating DNAP is able to displace an RNAP and subsequently the DNAP can re-initiate DNA synthesis using the RNA transcript. These findings reveal a novel pathway of replication re-initiation enabled by the participation of a replicative helicase.

## Results

**Non-replicating DNAP regulates helicase slippage and rate.** During leading-strand replication, DNA synthesis by an actively elongating DNAP has been shown to facilitate T7 helicase unwinding[21, 26, 27]. However, it is unclear if the non-replicating DNAP, which is disengaged from DNA synthesis, as could occur after a replisome encountering a lesion, still affects helicase unwinding. Therefore, we first addressed whether a non-replicating DNAP could still facilitate helicase unwinding. Previously, we discovered that T7 helicase, when powered by ATP alone, unwinds DNA rapidly but frequently slips, which is in contrast to its processive unwinding activity in the presence of dTTP[28]. During a slippage event, helicase loses its grip on the tracking ssDNA, slides backwards under the influence of the re-annealing DNA fork, and then regains its grip to resume unwinding (Fig. 1a). We thus examined T7 helicase's slippage and unwinding activities (rate and processivity) in the presence of non-replicating DNAP and 2 mM ATP. This nucleotide condition supports DNA unwinding but does not support DNA replication because dNTPs are missing. We employed a previously developed single-molecule optical trapping assay to measure T7 helicase unwinding of dsDNA[29]. Briefly, to mimic a stalled fork with a leading-strand gap, we generated a ssDNA region of approximately 900 nt in the leading strand near a fork junction by mechanically unzipping the dsDNA. Subsequent helicase unwinding of the junction led to an increase in the ssDNA length, allowing tracking of the helicase location (Supplementary Fig. 1a). Once helicase activity was detected, then helicase catalyzed unwinding was monitored under a constant force, which was not sufficient to mechanically unzip the fork junction (Fig. 1b). In the absence of DNAP and using only ATP as the fuel source, T7 helicase was found to frequently slip during dsDNA unwinding (Fig. 1b), consistent with our previous findings[28]. These slippage events led to a remarkable sawtooth pattern in an unwinding trace. The processivity, which is defined as the average distance between slips, was $400 \pm 50$ bp (mean ± s.d.) under 8 pN of force (Fig. 1d). Surprisingly, we found that in the presence of the non-replicating DNAP, T7 helicase unwound DNA processively without detectable slippage through the entire dsDNA available (~2500 bp) (Fig. 1c, d). Because previous studies showed that T7 DNAP's functional activities, such as processive synthesis and strong binding on the template, require the association of gp5 with the processivity factor trx[30], we investigated whether trx is required to prevent helicase slippage. We found that T7 helicase slipped in the presence of gp5 alone or trx alone but not when both were present, indicating that both gp5 and trx are necessary to prevent helicase slippage (Fig. 1d and Supplementary Fig. 2). For all subsequent experiments with DNAP, trx was also present.

Next, we examined the force dependency of stimulation of helicase unwinding by the non-replicating DNAP. At forces below 9 pN, helicase alone unwound with frequent slippage, and DNAP increased the helicase unwinding rates between slips, whereas this effect was negligible in a higher force region (Fig. 1e): Student's t-test $t(7) = -0.65$ (10 pN) and $t(6) = 0.87$ (12 pN), $p > 0.05$. A similar trend in unwinding rates was observed in experiments carried out in the presence of dTTP (Supplementary Fig. 3). This is also supported by our previous bulk study where T7 DNAP enhanced the catalytic efficiency of unwinding by T7 helicase in the absence of dNTPs (with dTTP alone as fuel for helicase activity)[26].

To examine whether the increase in helicase unwinding activities by a non-replicating DNAP is via their direct interactions, we utilized a mutant T7 helicase that lacks the 17 carboxyl-terminal amino acid residues ($\Delta C_t$) required for interaction with T7 DNAP[31, 32]. The $\Delta C_t$ mutant of T7 helicase

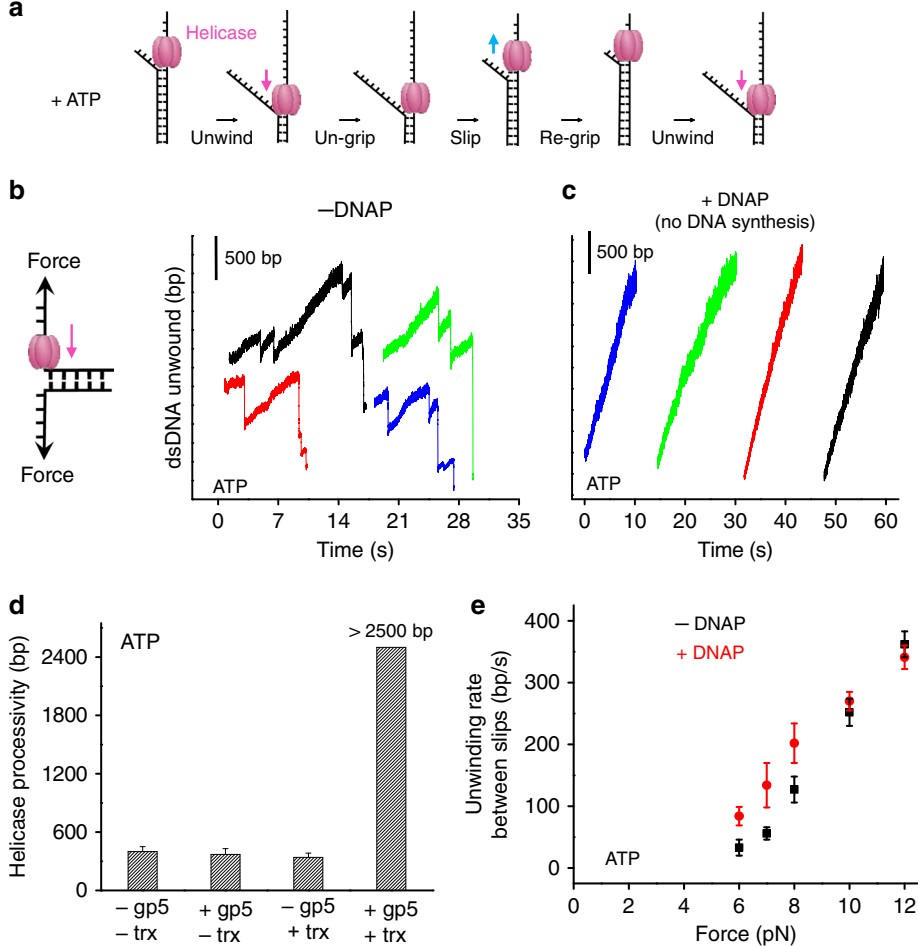

**Fig. 1** Non-replicating DNAP prevents helicase slippage and increases unwinding rates. **a** Cartoon illustrating T7 helicase unwinding and slippage behavior. The helicase unwinds, loses grip, slips, re-grips, and resumes unwinding. Arrows indicate the directions of helicase movements. **b**, **c** Representative traces showing the number of unwound base pairs by T7 helicase vs. time in the absence (28 traces in total) or presence (25 traces in total) of non-replicating DNAP, respectively. Experiments were conducted with 2 mM ATP under 8 pN force. For clarity, traces have been arbitrarily shifted along both axes. **d** Measured processivity of T7 helicase (mean distance between slippage events) in the absence and presence of gp5 and/or trx with 2 mM ATP under 8 pN external force. **e** T7 helicase unwinding rates between slips as a function of force in the absence or presence of the DNAP with 2 mM ATP. Unwinding rates were determined between slippage events (if any). Error bars represent standard deviations

has DNA unwinding activity, but does not form a stable complex with T7 DNAP[31]. We carried out similar unwinding experiments to those described above except with the $\Delta C_t$ mutant. We found that slippage occurred both in the presence and absence of a non-replicating DNAP. The non-replicating DNAP did not change the processivity or the unwinding rate of the mutant helicase (Supplementary Fig. 4), which is in stark contrast to the results observed with wt helicase (Fig. 1). Therefore, direct interactions between a helicase and a non-replicating DNAP are essential for slippage prevention and unwinding rate enhancement.

**Non-replicating DNAP and helicase localize to the fork.** Finding that a non-replicating DNAP facilitates helicase unwinding raises a question about the configuration of the helicase and the DNAP at the fork. If DNAP is able to bind to the leading strand while interacting with helicase on the lagging strand across the fork junction, then it will be poised for leading-strand synthesis once it acquires a primer (Fig. 2a). We, therefore, designed an experiment to investigate whether a non-replicating DNAP binds to the leading strand at the fork junction during helicase unwinding. First, in the presence of helicase, dTTP, and with or without DNAP, we mechanically unzipped approximately

900 bp of a 4100 bp dsDNA template (at a speed of 250 bp/s) (step 1). This created a ssDNA region for helicase loading and allowed subsequent translocation of the helicase to the fork junction. Helicase presence was monitored by DNA unwinding activity under a constant force of 9 pN (step 2). Once helicase unwound ~1000 bp, we rapidly unzipped the remaining DNA mechanically at 2000 bp/s (much faster than helicase unwinding rate) to detect any bound proteins across the fork junction (step 3). During this mechanical unzipping step, a force-rise significantly above the naked DNA baseline served as a sensitive detector for the presence of a bound protein across of the fork junction[33–36].

In the absence of DNAP, no significant force-rise above the naked DNA baseline was detected during step 3 (Fig. 2b), suggesting that helicase translocates on the lagging-strand and unwinds DNA while having minimal interactions with the leading strand. In contrast, in the presence of non-replicating DNAP, in step 3, about 75% of the traces (31 traces in total) exhibited a force-rise significantly above the naked DNA baseline, followed by a return of the unzipping force to the naked DNA baseline (Fig. 2c). This demonstrates that a protein or protein complex was located across the fork junction in those molecules. The peak of the force-rise was $26 \pm 1$ pN (mean ± s.d.)

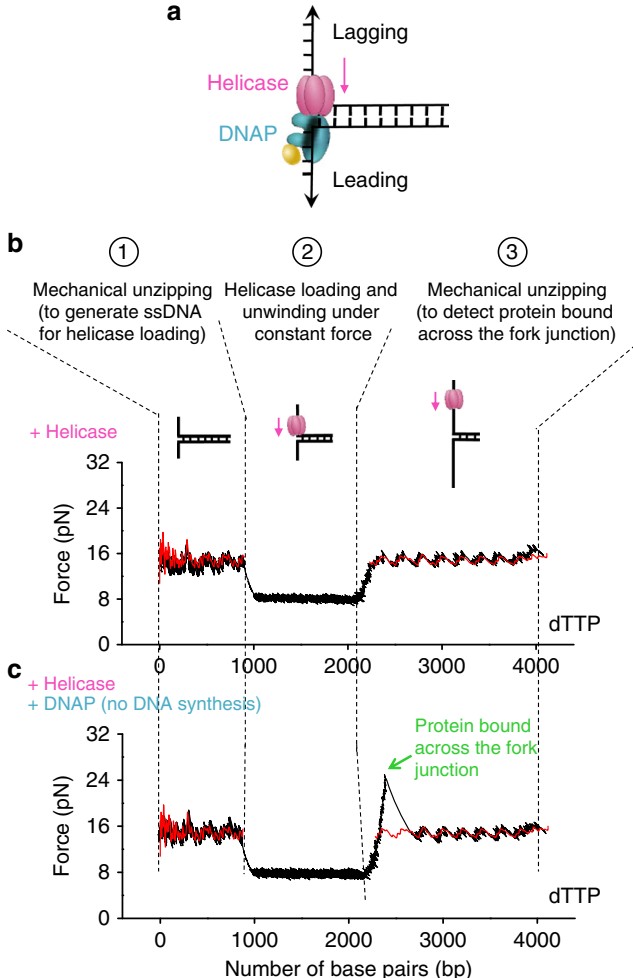

**Fig. 2** Non-replicating DNAP localizes to the fork with the helicase. **a** A cartoon illustrating helicase-DNAP coupling. A non-replicating DNAP directly interacts with an unwinding helicase. The helicase–DNAP complex is juxtaposed crossed the fork junction. **b**, **c** Representative traces showing the force vs. number of base pairs unzipped/unwound in presence of helicase and 2 mM dTTP, either without or with the presence of non-replicating DNAP, respectively. The red curves correspond to unzipping naked DNA. The green arrow indicates a force peak above the naked DNA baseline

(Supplementary Fig. 5) and the naked DNA baseline averaged to ~15 pN. The absence of a force-rise under the helicase only condition (Fig. 2a) precludes the possibility that this force-rise is due to interactions of helicase alone with the DNA fork junction. The force-rise was also absent in a control experiment with DNAP and $\Delta C_t$ helicase (Supplementary Fig. 6), suggesting that direct interactions between DNAP and helicase are essential. The force-rise is thus attributed to the helicase and DNAP interactions across the fork junction. For the 25% of traces that did not show detectable force-rise above the naked DNA baseline, it is possible that DNAP was not present at the fork junction. Because we observed a complete lack of slippage during helicase unwinding in the presence of DNAP (Fig. 1), we favor the possibility that DNAP was present at the fork, but the DNAP–helicase or DNAP–DNA interactions were transiently lost at the moment of detection. Ultimately, we conclude that a non-replicating DNAP directly interacts with an unwinding helicase across the fork junction (Fig. 2a).

**Helicase and DNAP displace RNAP and re-initiate replication.** A non-replicating DNAP localized at a fork junction via an unwinding helicase is well poised for re-initiating replication upon primer acquisition. This may take place during a co-directional collision with a transcription elongation complex (TEC) if a helicase-DNAP complex is able to displace the RNAP and allow DNAP access to the RNA transcript[7]. Although we have demonstrated that a non-replicating DNAP enhances helicase's motor ability (Fig. 1), it is unclear whether the non-replicating helicase–DNAP complex is capable of overcoming the TEC barrier.

To investigate the outcome of such a collision, we developed a real-time assay to monitor the co-directional progression of a helicase with a non-replicating DNAP through a stalled *E. coli* TEC (Supplementary Fig. 1b). First, to differentiate between unwinding without and with DNA replication, we carried out two control experiments to characterize the rates of extension under a constant unzipping force of 5 pN (Supplementary Fig. 1b) in a reaction buffer containing all four dNTPs, T7 helicase, and T7 DNAP. In the first experiment, the leading strand was provided with a DNA primer containing an inverted dT at its 3′ end that does not support DNA synthesis even in the presence of all dNTPs. The DNA length increased at a rate of 34 ± 20 nm/s (mean ± s.d) (Fig. 3a). In the second experiment, the leading strand was provided with a DNA primer from which T7 DNAP could synthesize, and the DNA length increased at a rate of 91 ± 18 nm/s (mean ± s.d) (Fig. 3a). Thus, unwinding rate is slower with a non-replicating DNAP and becomes faster with a replicating DNAP. A transition from a slow to this faster rate serves as a clear indicator of the onset of DNA synthesis.

To directly examine the consequences of a collision between the helicase/non-replicating DNAP and a TEC, we used a parental DNA template that contained an inverted dT primer and a co-directional TEC stalled at +20 nt from its promoter (Fig. 3b). There should only be unwinding without DNA synthesis prior to their collision. Consistent with this, all traces exhibited an expected slow DNA length increase rate before collision (Fig. 3c, e). Upon collision, several types of behavior were observed. In 88% of 68 traces, DNA length continued to increase at a rate similar to that before collision (Fig. 3c, e), consistent with unwinding without DNA synthesis. Within this 88%, 43% of the traces showed a detectable pause at the expected RNAP position (Fig. 3c), suggesting that the helicase-DNAP complex was able to overcome the TEC roadblock, but the complex moved forward without DNA synthesis; while the rest did not show any detectable pausing, in part due to absence of a TEC during the initial TEC formation (Methods). Interestingly, the remaining 12% of traces paused transiently at the expected RNAP position and then transitioned to an increased rate consistent with that of unwinding with leading-strand synthesis (Fig. 3d, e). For these traces, the helicase–DNAP complex was indeed able to overcome the TEC roadblock, and the DNAP was then able to carry out DNA synthesis. Normalizing against the initial TEC formation efficiency, the replication re-initiation efficiency is ~15%. T7 gp4 also contains a primase domain, but this cannot be the source of primers in our experiments due to lack of priming nucleotides, ATP and CTP.

We previously demonstrated that DNAP provided with a DNA primer that can be extended, is unable to efficiently replicate the fork, and the double-stranded DNA represents a major barrier for the DNAP under low unzipping force[15]. In contrast, under a similar condition (5 pN and 0.5 mM dNTP), T7 helicase alone is able to unwind and displace RNAP (Supplementary Fig. 7). Thus, T7 helicase is primarily responsible for displacing RNAP and this displacement is facilitated further by synergistic interactions of helicase with a non-replicating DNAP. The synergistic

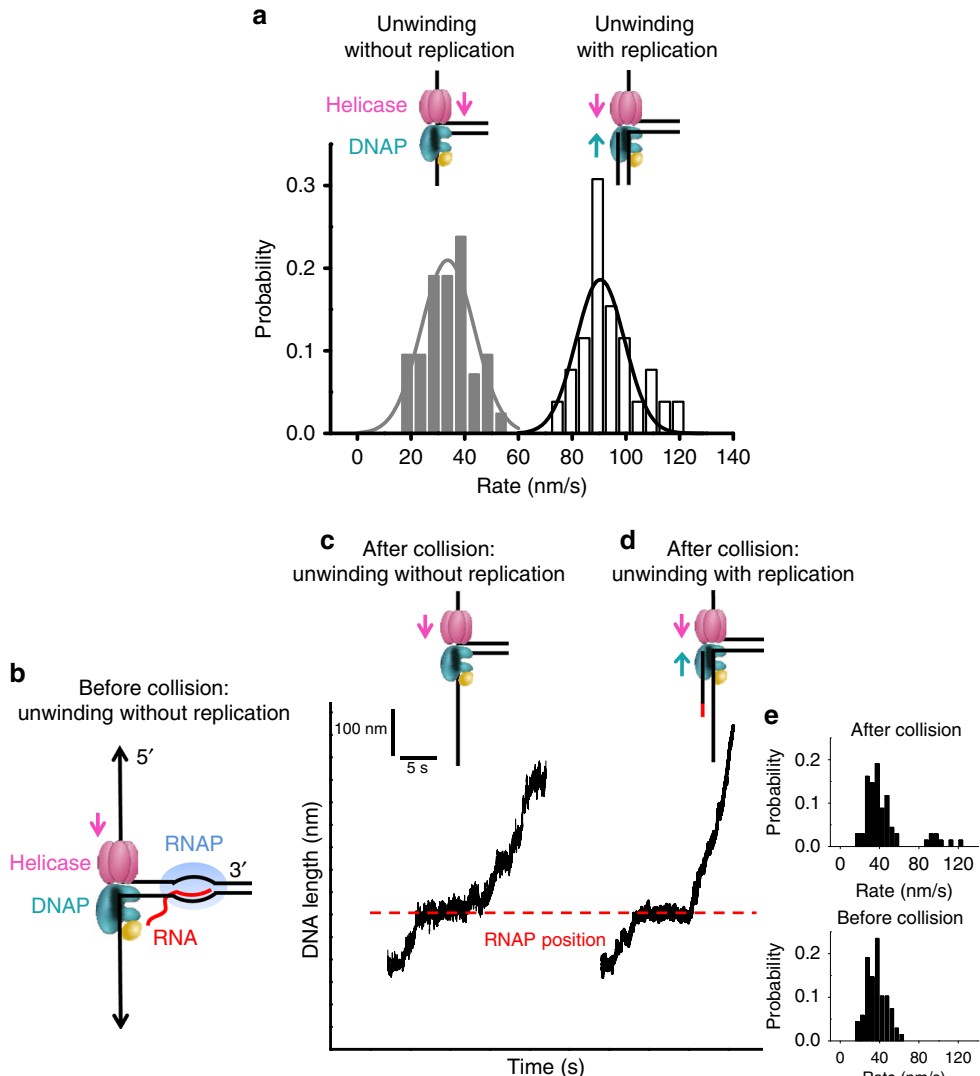

**Fig. 3** Single-molecule experiments on helicase and non-replicating DNAP collision with a TEC. **a** Distributions of DNA length increase rates of helicase unwinding with or without replication in 0.5 mM dNTP (each) under 5 pN of force. **b** Experimental design. Left panel shows a cartoon of *E. coli* TEC that was stalled at +20 nt from the promoter while a helicase with a non-replicating DNAP encountered the TEC co-directionally. **c** A representative trace of unwinding without replication after collision. Experiments were carried out in the presence of helicase, DNAP, and 0.5 mM dNTP (each) under 5 pN of force. The dotted line indicates the expected stalled TEC position. The cartoon on the top illustrates replication status after the collision with RNAP. **d** A representative trace of unwinding with replication after collision. Same experimental conditions were used as in **c**. For clarity, traces have been shifted along the time axis. **e** Distributions of DNA length increase rates before (upper panel) and after (lower panel) the collision with RNAP

interactions greatly enhance helicase unwinding activities, while holding the DNAP close to the unwinding fork to re-initiate leading-strand synthesis. We conclude that helicase in association with a non-replicating DNAP forms a strong motor complex at the fork, capable of displacing an RNAP. As a result, DNAP can gain access to the RNA transcript and use it as a primer for replication initiation.

**Ensemble studies support helicase's role in re-initiation**. To exclude the possibility that the observed replication re-initiation was a result of the applied force on the ssDNA under our single molecule conditions, we carried out corresponding ensemble experiments in the absence of any externally applied force. For these experiments, the replication fork substrate contained a stalled, co-directional T7 TEC on the parental dsDNA (Fig. 4a and Supplementary Fig. 8). As expected, the TEC was able to make run-off products with rNTPs (Supplementary Fig. 8), and T7 DNAP alone was able to efficiently extend the RNA primer on

ssDNA template and make run-off products in the presence of dNTPs (Fig. 4c). On the fork substrate containing a stalled TEC, we observed run-off products in the presence of dNTPs only when both helicase and DNAP were present. We quantified that ~14% of the RNA had fully extended to the end of 10 min (Fig. 4a, d). A very small amount of run-off product (~1.2% at the end of 10 min) was observed in the absence of helicase on forked TEC (Fig. 4a). In all lanes, we observed short products that migrated close to the 10-mer DNA markers. Evidently, TEC can use dNTPs as substrates but incorporates only up to three dNMPs[37] as shown in experiments with the fluorescein-labeled primer (Supplementary Fig. 9). These experiments in combination indicate that the run-off products result from replication re-initiation by T7 DNAP from the RNA primer in TEC, and the helicase plays an important role in replication re-initiation. As additional control experiments, we replaced the fork substrate with a blunt-end dsDNA, which T7 helicase cannot load onto and is thus unable to unwind (Fig. 4b)[38]. As with the forked substrate,

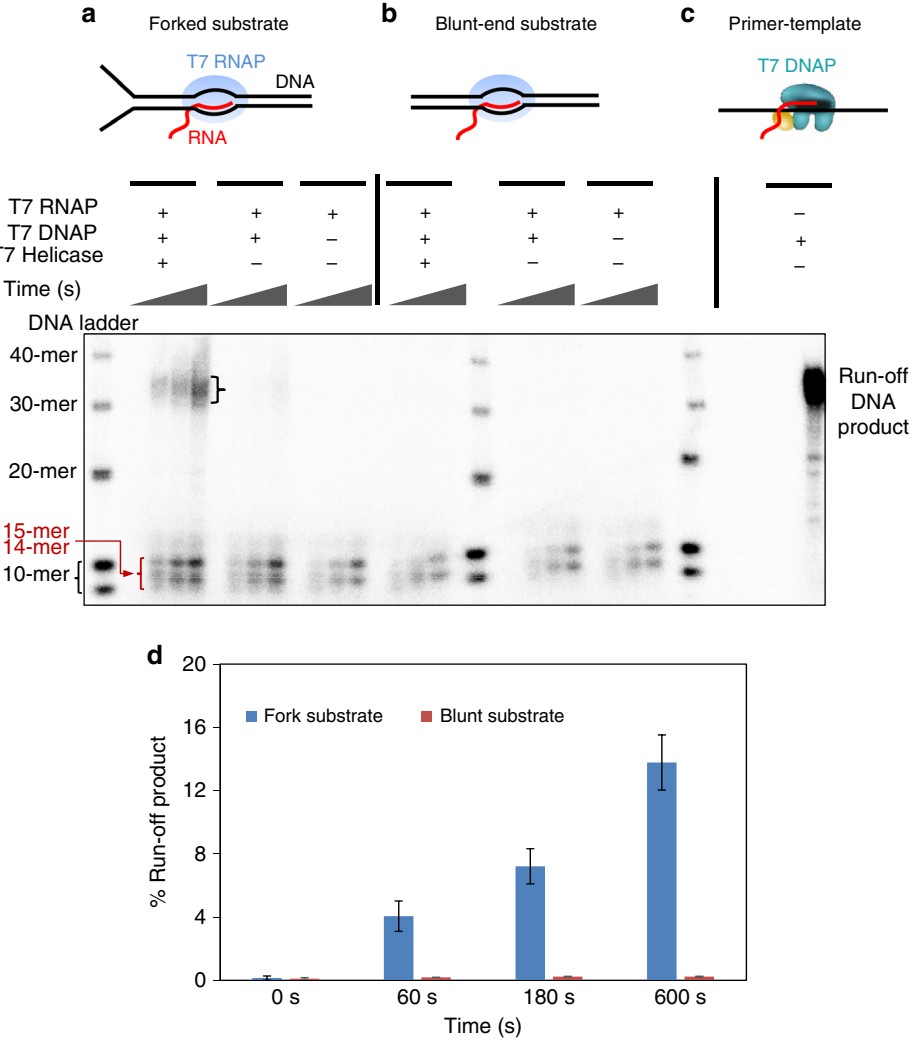

**Fig. 4** Bulk experiments on helicase and non-replicating DNAP collision with a TEC. **a**, **b** Experiments on fork DNA substrate (left) and blunt DNA substrate (right) respectively. The parental DNA contained a stalled T7 RNAP and experiments were carried out with dNTP mixture spiked with [α-$^{32}$P]-dGTP. For each experiment, reactions were quenched at four time points (0, 60, 180, and 600 s). Samples were mixed with formamide and bromophenol blue dye and heated at 95 °C for 5 min before loading on 12% acrylamide/6M urea sequencing gels. Sequencing gels show the kinetics of the RNA primer extension on either the fork substrate or the blunt substrate. The run-off DNA product is 38-nt long. The products running close to the 10-nt DNA markers are 14-mer and 15-mer resulting from dNTPs addition by T7 RNAP to the 12-mer RNA primer. **c** Replication reaction performed with just the primer annealed to the template. Experiment was carried out with dNTP mixture spiked with [α-$^{32}$P]-dGTP and T7 DNAP. Reaction was quenched at 600 s. The replication product was used as a control for quantitating the % run-off DNA products obtained in **a** and **b**. **d** Percentage run-off product estimated from **a** and **b**

the blunt dsDNA template contained a stalled TEC (Supplementary Fig. 8), and the TEC was able to use dNTPs to extend the RNA by a few nucleotides. However, fully extended primers were not detected on this blunt substrate with helicase and DNAP (Fig. 4b, d). Thus helicase unwinding is critical in enabling re-initiation.

These results are in agreement with our single-molecule findings that a non-replicating DNAP in conjunction with an unwinding helicase can utilize an RNA primer from a co-directional TEC and subsequently carry out continuous DNA synthesis. These data also reinforce the conclusion that helicase is essential in assisting the DNAP in re-initiating the leading-strand synthesis.

## Discussion
Although the long-established role of replicative helicases is to catalyze strand separation, emerging evidence now supports the notion that their functions in replication are much broader[39, 40].

Our findings here elucidate a novel role of T7 helicase in enabling replication re-initiation (Fig. 5). We show that during DNA unwinding, helicase strongly interacts with a non-replicating DNAP on the leading strand. The two proteins form a complex across the fork junction, and the interactions between them permit the non-replicating DNAP to travel alongside the unwinding helicase. The presence of the DNAP at the fork junction increases helicase's unwinding rate and processivity by preventing helicase slippage. Consequently, processive unwinding by the helicase associated with a non-replicating DNAP leads to TEC disruption during a co-directional collision with transcription, exposing the RNA transcript. Then the DNAP is able to pick up the RNA and use it as a primer to initiate DNA synthesis. Although previous work showed that a *replicating* replisome involving a T4 or *E. coli* leading-strand DNAP may overcome a TEC barrier[17, 41, 42], this work shows that even if the T7 DNAP is not replicating, T7 helicase can enhance its capacity to overcome barriers and re-initiating replication. During the early phase of T7 transcription, *E. coli* RNAP transcribes T7 genes through specific

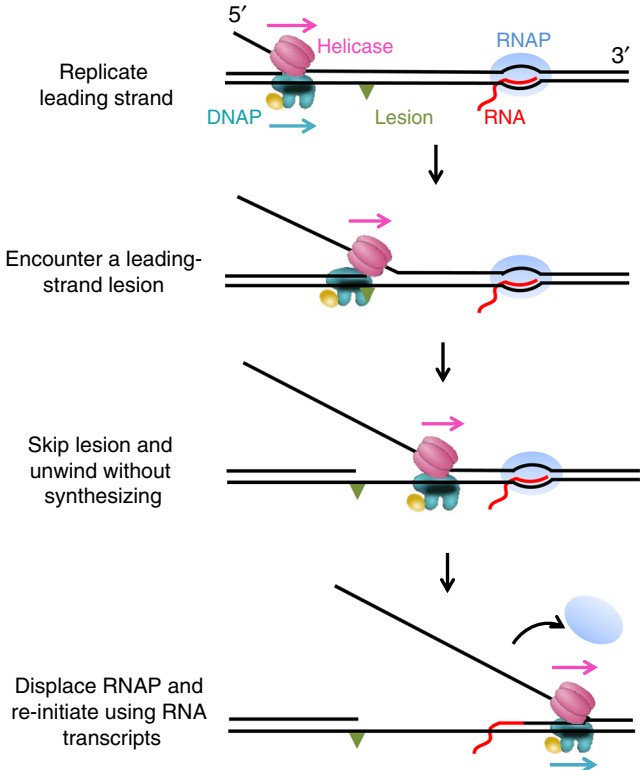

**Fig. 5** Proposed T7 replication re-initiation model. Cartoons illustrate a proposed model for T7 replication re-initiation. The replisome here consists of leading-strand DNAP and helicase. When the replisome encounters a leading-strand lesion, helicase may continue to unwind processively via association with a non-replicating DNAP. The two proteins form a complex clamping crossed the fork junction, poising the DNAP for replication re-initiation. After collision with a TEC, the helicase-DNAP displaces the RNAP and the DNAP then uses the RNA as a primer to re-initiate the replication

promoters[43, 44]. Thus, in addition to encounters of the T7 replisome with T7 RNAP, there is a possibility of encounters with the *E. coli* RNAP. Our study shows that both T7 and *E. coli* TECs can support T7 replication re-initiation, indicating that protein-specific interactions are not required.

The non-replicating DNAP stimulates the unwinding activity of the helicase by direct interactions. A unique feature of T7 replisome is that T7 helicase and DNAP interact physically through several modes[45, 46], confirmed by recent structural studies[32, 47]. The acidic C-terminal tail of helicase interacts with a front "basic-patch" at the front end of the DNAP facing the fork junction and the "basic-patch" in the thioredoxin-binding domain of DNAP. The interaction with the front basic patch is important for loading and the thioredoxin-binding basic patch is important for polymerase exchange. These interactions increase the processivity of the replisome from 5 to 17 kbp by capturing DNAP that transiently dissociates[48]. In addition, we previously demonstrated that these interactions ensure that a T7 replisome directly synthesizes through a DNA lesion[15]. We propose that the non-replicating DNAP and helicase bound to opposite DNA strands act synergistically. When helicase loses grip on DNA, the DNA-bound non-replicating DNAP holds helicase on the DNA template via DNAP–helicase interactions, thus preventing helicase slippage or dissociation. Furthermore, the non-replicating DNAP at the fork junction is able to relieve some of the regression pressure from fork junction reannealing to increase the DNA unwinding rates.

This work also provides insights into the transcription-initiated replication in T7, T4, *E. coli* ColE1 plasmid, mitochondrial DNA replication, and origin-independent replication initiation in eukaryotes, whose mechanisms remain elusive[49]. In transcription initiated replication of T7, it was found that helicase enhances DNAP's acquisition of RNA primer from T7 RNAP[50–53]. In this process, DNAP must be positioned in close vicinity to the T7 RNAP. Thus, DNAP's association with helicase may be beneficial if the two form a strong complex on the DNA that can displace the RNAP and pick up the primer from the R-loop.

Our work supports that T7 helicase participates in the assembly of the replication machinery at the fork and helps resolve replication conflicts with roadblocks on the DNA. Thus helicase has broad functionalities and unexpected roles in assuring processive DNA replication.

## Methods

**Protein and DNA preparations**. Untagged gp4A′ (wild-type helicase) and delta C gp4A′ mutant (ΔCt) were overexpressed in the BL21 DE3 cell line[31, 54]. The cells were lysed by three freeze-thaw cycles in 20 mM phosphate buffer pH 7.4, with 50 mM NaCl, 10% glycerol, 2 mM DTT, 2 mM beta mercapto-ethanol and 1 or 2.5 mM EDTA in the presence of 0.2 mg/ml lysozyme. Polyethyleneimine precipitation was carried out by increasing the salt concentration to 0.5 M. The supernatant was precipitated in 70% ammonium sulfate and purified by Phosphocellulose (P11 resin) followed by DEAE Sepharose column chromatography. *E. coli* thioredoxin (trx) was purchased from Sigma-Aldrich (St. Louis, MO). *Escherichia coli* RNAP was expressed and purified as previously described at a low level in wild-type *E. coli* strains (WT, DH5α) to yield RNAPs[55]. *Escherichia coli* RNAPs were purified to homogeneity by using a modification of the method of Burgess and Jendrisak to include chromatography on nickel agarose[56]. T7 RNAP was overexpressed in *E. coli* strain BL21/pAR1219 and purified using three chromatographic columns consisting of SP-Sephadex, CM-Sephadex, and DEAD- Sephacel[57, 58]. The purified enzyme was dialyzed against buffer (20 mM sodium phosphate, pH 7.7, 1 mM Na₃-EDTA, and 1 Mm DTT) containing 100 mM NaCl and 50% (v/v) glycerol, and stored at −70 °C. Wild-type T7 gp5 and exo-gp5 (D5A, D7A) were purified as previously described[54, 59].

The DNA template for helicase unwinding and unzipping consisted of a 1.1 kbp anchoring segment and a 4.1 kbp unwinding segment (Supplementary Fig. 1a)[60]. In brief, the anchoring segment was amplified from plasmid pRL574[61] and the unwinding segment was derived from 17 pseudo-repeats (or 17mer) of the 5s rRNA sequence[62]. The final product was produced by ligating the two segments in a 1:1 molar ratio. The DNA template for the helicase-DNAP coupled replication initiation assay consisted of three pieces: two arms and a trunk (Supplementary Fig. 1b). Arm 1 (1129 bp) was PCR amplified from plasmid pRL574 using a digoxigenin-labeled primer. The resulting DNA fragment was digested with BstXI (NEB, Ipswich, MA) to create an overhang and was subsequently annealed to a short DNA with a complementary overhang formed by adapter 1 (5′-/phos/GCA GTA CCG AGC TCA TCC AAT TCT ACA TGC CGC-3′) and adapter 2 (5′-/phos/GCC TTG CAC GTG ATT ACG AGA TAT CGA TGA TTG CG GCG GCA TGT AGA ATT GGA TGA GCT CGG TAC TGC ATCG-3′). Arm 2 (2013 bp) was PCR amplified from plasmid pBR322 (NEB, Ipswich, MA) using a biotin-labeled primer. The resulting DNA fragment was digested with BstEII (NEB, Ipswich, MA) to create an overhang and was subsequently annealed to adapter 3 (5′-/phos/GTAAC CTG TAC AGT GTA TAG AAT GAC GTA ACG CGC AAT CAT CGA TAT CTC GTA ATC ACG TGC AAG GC CTA-3′). The adapter 3 from arm 2 and the adapter 2 from arm 1 were partially complementary to each other and were annealed to create a short ~30-bp trunk with a 3-bp overhang for the trunk ligation. The 1.5 kbp trunk containing the T7A1 promoter was amplified from plasmid pRL574. The final product was produced by ligating the arms with the trunk at 1:4 ratio.

**Single-molecule assays**. Sample chamber preparation was similar to that previously described[15, 28, 60]. Briefly, DNA tethers were formed by first non-specifically coating a sample chamber surface with anti-digoxigenin (Roche, Indianapolis, IN), which binds nonspecifically to the coverglass surface, followed by incubation with digoxigenin-tagged DNA. Streptavidin-coated 0.48 mm polystyrene microspheres (Polysciences, Warrington, PA) were then added to the chamber. DNAP was assembled by adding 10 μM of exo-gp5 in 50 μM *E. coli* trx and incubating at room temperature for 5 min. The helicase and DNAP were prepared as follows: first, 100 nM of the appropriate helicase hexamer was incubated for 20 min in the replication buffer on ice; then 100 nM of the appropriate DNAP was added, and the solution was incubated for 10 min at room temperature. This solution was then further diluted to obtain the final experimental concentrations of helicase and DNAP, nucleotides and MgCl₂. The resulting solution was flowed into the chamber just prior to data acquisition. The unwinding and replication buffer consisted of 50 mM Tris–HCl pH 7.5, 40 mM NaCl, 10% glycerol, 1.5 mM EDTA, 2 mM DTT and

dNTPs at the concentrations specified in the figure legends, and $MgCl_2$ at a concentration of 2.5 mM in excess of the total nucleotide concentration. Paused transcription complexes were formed by incubating 2 nM E. coli RNAP, 0.4 nM DNA template and 1 mM ApU, and 1 mM ATP/CTP/GTP in transcription buffer for 30 min at 37 °C[34].

Experiments were conducted in a climate-controlled room at a temperature of 23.3 °C, but owing to local laser trap heating the temperature increased slightly to $25 \pm 1$ °C[63]. Each experiment was conducted in the following steps. First, several hundred base pairs of dsDNA were mechanically unzipped, at a constant velocity of 1400 bp/s (helicase unwinding assay) or 250 bp/s (DNAP binding detection assay), to produce a ssDNA loading region for helicase. Second, DNA length was maintained until a force drop below a preset value, indicating helicase unwinding of the DNA fork. Third, a constant force was maintained at this preset value via computer-controlled feedback, while helicase unwound the dsDNA. In the unzipping force analysis of helicase and DNAP association, after the detection of helicase loading and unwinding, the remaining dsDNA was mechanically unzipped at an extremely fast velocity of 2000 bp/s to probe the potential interactions at the fork.

**Data collection and analysis**. Data were low-pass filtered to 5 kHz and digitized at 12 kHz, then were further averaged to 110 Hz. The acquired data signals were converted into force and DNA extension as previously described[29]. In the helicase-unwinding studies, one separated base pair generated two nucleotides of ssDNA. Accordingly, real-time DNA extension in nm was further converted into the number of base pairs unwound based on the elastic parameters of ssDNA under our experimental conditions. To improve positional accuracy and precision, the data were then aligned to a theoretical unzipping curve for the mechanically unzipped section of the DNA. Helicase unwinding rates were determined as previously described[29]. For the helicase–DNAP coupled replication initiation assay, we had to determine whether the movement after the RNAP was due to helicase alone or helicase-coupled DNAP synthesis and this was more readily achieved by directly measuring the DNA length increase rate in nm/s. Therefore, we presented data as the DNA length in nm and rates in nm/s. The position of a paused TEC was known from the DNA template design (753 bp from the initial fork). Its position in nm showed in Fig. 3 was determined by converting bp to nm.

**Ensemble assays**. For experiments described in Fig. 4, fork substrate and blunt substrate (sequences provided in Supplementary Fig. 8a) were annealed by mixing template, non-template, and 5′-fluorescein labeled RNA primer in 1.25:1.5:1 ratio in Tris–HCl pH 7.5, 1 mM EDTA, 50 mM NaCl buffer, incubating the mixture at 95 °C for 2 min and gradually cooling it to 20 °C. The TEC was assembled by mixing the substrate (200 nM) with T7 RNAP (1100 nM), T7 helicase (220 nM), and dTTP (1 mM) at 18 °C for 60 min in buffer containing Tris–HCl pH 7.5, 40 mM NaCl, 10% glycerol, 2 mM EDTA and 2 mM DTT. Assembled TEC was then incubated with exo− or exo+ T7 gp5 (220 nM) and thioredoxin (1100 nM) for 60 min. Reactions were initiated by adding the rest of the dNTPs (200 μM each, spiked with [α-$^{32}$P]-dGTP) and $MgCl_2$ (final concentration 5 mM in the reaction). After preset time intervals (0, 60, 180, and 600 s), the reactions were stopped with EDTA (0.15 M), mixed with formamide and bromophenol blue dye, heated at 95 °C for 5 min and loaded on a 12% polyacrylamide/6M urea sequencing gels. Gels were exposed to phosphor screens, and the screens were scanned with Typhoon FLA 9500 scanner (GE Healthcare). Replication reaction was also performed with just the primer annealed to the template. The replication product of this reaction at 600 s was used as a control for quantitating the percent run-off DNA products. For markers, 10-bp dsDNA ladder from Invitrogen (Life Technologies) was used. On the denaturing gels, the two strands of the 10mer were resolved into a double band likely due to the slight sequence difference between the two strands.

**Data availability**. The data that support the findings of this study are available from the corresponding authors upon request.

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

## Acknowledgements

We thank Dr. Robert M. Fulbright Jr. for purification of the *E. coli* RNA polymerase, Dr. Manjula Pandey for purification of T7 helicase and T7 DNAP, and Dr. Shanna Moore and the Sun laboratory for critical reading of the manuscript. We wish to acknowledge support from National Key Research and Development Program of China (2016YFA0500902 and 2017YFA0106700 to B.S.), Shanghai Pujiang Program (16PJ1406900 to B.S.), National Institutes of Health grant (R35GM118086 to S.S.P.), Howard Hughes Medical Institute (to M.D.W.), and National Science Foundation grant (MCB-1517764 to M.D.W.).

## Author contributions

B.S. and M.D.W. designed the single-molecule experiments. B.S. carried out the single-molecule experiments and analyzed and interpreted single-molecule data. J.T.I. maintained and upgraded the optical trapping setup. A.S., S.S., and S.S.P. designed the ensemble replication assays. S.S and A.S. performed ensemble experiments. B.S. and M. D.W. drafted the manuscript. All authors contributed to revisions of the manuscript and intellectual discussions.

## Additional information

**Competing interests:** The authors declare no competing interests.

