## [Peer Review File · Nature Communications]

Reviewers' Comments:

Reviewer #1:

Remarks to the Author:

This manuscript by Sun, Patel, Wang and colleagues provides interesting single molecule data showing 1) that DNAP interacting with the helicase can increase the unwinding rate and processivity and that 2) this interaction can be used to facilitate replication restart by displacing a head on transcription complex and reengaging the mRNA transcript to continue replication. Both of these results are exciting a novel and should provide a basis for testing similar features in higher organisms. The manuscript is well written and sound and most of my comments below are of a genuine curiosity. However, there are certain aspect that need to be clarified to strengthen this manuscript.

- 1) Intro, what is the predicted frequency of a replication-TEC headon collision? How often would this be expected say in T7 or bacteria. Provide a reference for significance.
- 2) "non-replicating DNAP" needs to be defined from the start. I believe you are using cat- mutant of gp5, but this should be explicitly stated. What happens when experiment in Fig 1c is performed with WT gp5 but in the absence of dNTPs. I imagine the result is the same? Also later you use "nonreplicating" when describing an inverted dT. You should be specific in each case.
- 3) This is sentence seems wrong? "Once helicase activity was detected, then helicase catalyzed unwinding was monitored under a constant force, which was not sufficient to mechanically unwind the fork junction without a helicase (Fig. 1b)." without a DNAP?
- 4) what was the difference in rate at 8pN between Fig 1b and c? Is it the same as in Fig 3a?
- 5) Page 7, Fig 2c, Is the height of the force peak reproducible in those 75% of traces. Can't the magnitude be used to quantify the strength of the interaction of helicase-DNAP with DNA?
- 6) Page 8, there does not seem to be proof for "DNAP-helicase interactions were transiently lost at the moment of detection". Couldn't it just as easily be that DNAP lost contact with DNA, but remained bound to helicase in those cases. The probability seems to low that 15% of the molecules were lost just as you were measuring force change?
- 7) Page 9, would be nice to see an example without any detecable pausing.
- 8) page, 9, any correlation of length of pause with increased rate of 2nd rate of synthesis. For example is a pause needed to get a change in rate correlating with TC-restart?
- 7)Page 10, Fig 4a, lanes with RNAP and DNA but no helicase. Looks like there is a small amount of runoff product here. But you say in text, "However, no run-off products were observed with dNTPs on a fork substrate when helicase was absent (Fig. 4a)."
- 8) Fig 4, Would have been nice to have a lane with RNA primer only to more clearly see the products you describe (i.e. TEC +dNTP +1 and +2 products). Also why is there 2 markers corresponding to 10mer? This is confusing. I realize that RNA and DNA can run differently, but we need to be clear what we are looking at?
- 9) Top of page 11, "we estimated that ~ 14% of the RNA had fully extended" Estimated or quantified?
- 10) WT T7 gp5 should also be included in methods.

Reviewer #2:

Remarks to the Author:

The manuscript entitled "Helicase promotes replication re-reinitiation from and RNA transcript " by Sun et.al. is an exciting and impactful paper by leading experts in the field, continuing a very strong collaboration between these two groups. The paper should draw interest from a wide variety of labs in diverse fields of replication, transcription and repair. The data are quite nice, and the important conclusions are supported by the data. I have a few comments below that might improve the information content for the readership, but overall the manuscript is quite acceptable in my opinion.

Specific comments:

1. It seems that the Intro of the paper could be a bit more informative about primase induced restart, especially considering the T7 helicase is the T7 primase and I don't think this was mentioned in the paper. One might presume the gp4 helicase-primase can prime both strands, as shown previously in the E coli and eukaryotic systems – but I'm not sure whether this has been demonstrated directly for T7 gp4. Maybe T7 gp4 simply can't do this? If this line of thought, it would be easy and useful to explain to readers the data/observations that eliminate T7 gp4 primase as a source of RNA priming in the experiments.

2. The single molecule experiments use E. coli RNAP, but considering the genome is T7 and encodes its own RNAP, could the authors comment on whether T7 replisome will be expected to encounter E coli RNAP transcripts during T7 replication? Possibly the mRNA takeover does not require protein-specificity of interaction – and this reviewer actually finds that interesting and plausible. But however the authors would like to comment on it is fine, but seems noteworthy to point out.

3. The ensemble work uses T7 RNAP and it would be useful to explain to readers the rationale for using the different RNAP's for single molecule vs ensemble experiments. I presume each RNAP has characteristics that make the use of one better for single molecule work and the other for ensemble studies?

4. It would be informative to mention the “n” of experiments performed for the various experiments, perhaps just a text note in the figures themselves – but otherwise in the legends or main text.

Reviewer #3:

Remarks to the Author:

“Helicase Promotes Replication Re-initiation from an RNA Transcript” by Sun et al. reports on new single-molecule optical tweezers experiments—supplemented by bulk biochemical assays—on T7 helicase and DNA polymerase (DNAP), essential components of the T7 replication machinery. In this work, the authors claim that a non-replicating T7 DNAP can enhance the activity of T7 helicase by reducing its slippage, leading to a faster and more processive unwinding. This enhancement is shown to be mediated through direct interactions between helicase and DNAP, with the complex formed across both strands of the DNA duplex at the DNA fork. Furthermore, the authors show that the helicase-DNAP complex can displace a stalled RNA polymerase (RNAP) and use a nascent RNA transcript as a primer to re-initiate DNA replication, at an efficiency of approximately 15%.

Overall, this is a very interesting manuscript that sheds new light on the synergy between replicative helicases and polymerases, and provides more evidence for the broader role these two proteins may play. Generally the approach appears sound and several of the experiments are elegantly conceived. Also, bulk assays nicely complement the findings on replication re-initiation. The findings should be of broad interest to the scientific community working on replication and on molecular machines. Overall, the work is publishable, but there are several important points (outlined below) that need to be addressed before a final recommendation can be made.

Main questions:

Overall it was unclear how the authors ensured that DNAP was non-replicative in the measurements in Fig. 1, 2, and part of 3. Are dNTPs withheld in these measurements to block synthesis? If so this was not stated in the text. Or, is this claim made solely based on the lack of primer to initiate replication? If the latter, is there any effect of dNTPs on the observed DNAP-mediated enhancement in helicase activity? According to their model, the helicase is essentially

pulling DNAP along the DNA; is DNAP then able to bind and incorporate dNTPs?

On a related point, what was the concentration used for each dNTP in the measurements? In the methods section, it is stated that the "concentrations [are] specified in the text", but they are not.

In general there is a lack of discussion on a potential molecular mechanism of enhancement. Could the authors speculate how the non-replicative DNAP acts as a barrier against helicase slippage? What prevents DNAP itself from slipping backward when it is not elongating a DNA strand, i.e. when there is no physical barrier preventing it from moving backward?

In the text, the authors state that they "anticipate that in the absence of force, helicase's unwinding activities may also be stimulated by the presence of a non-replicating DNAP." As shown in Fig. 1e, the effect of force on unwinding rate is significant, with no unwinding detected below 4 pN (presumably, the signal is too low to detect below this force). Although the authors show zero-force, bulk measurements of DNA synthesis, no bulk unwinding assays are shown demonstrating DNAP enhancement at zero force. The authors should address this.

In Fig. 1e, the authors claim that DNAP increases the unwinding rate of T7 helicase for forces < 9 pN but has no effect for forces > 9 pN. The authors should make clear how statistically significant the difference in rates (with and without DNAP) is below 9 pN, as this statement was not particularly convincing. Are the error bars shown in the figure standard error of the mean or standard deviation? This should be made clear in the figure caption, and a statistical analysis provided to assess the significance of any observed differences in rates.

Minor points:

In the section "Helicase assists a non-replicating DNAP in displacing RNAP and re-initiating replication", second paragraph, the authors state that "the extension was monitored under a low constant unzipping force". The force or force range should be stated explicitly.

In the section "Non-replicating DNAP with a helicase localizes to the fork junction" the authors describe experiments in which the DNAP-helicase complexes are ruptured by force (Fig. 2). For complexes that were presumably broken by applying force, did the authors observe resumption of unwinding after the force was lowered again? If so, was slippage observed, pointing to the helicase no longer making physical contact with DNAP?

In the measurements of RNAP displacement by the DNAP-helicase complex, is it clear whether DNAP or helicase are displacing RNAP? Is it known whether T7 helicase alone or replicating DNAP alone can displace RNAP or is the synergy of the DNAP-helicase complex required?

Response to Reviewers

Manuscript #	NCOMMS-17-33892
Title	Helicase Promotes Replication Re-initiation from an RNA Transcript
Corresponding Authors	Michelle D. Wang (Cornell University) & Bo Sun (ShanghaiTech University)
Contributing Authors	Bo Sun, Anupam Singh, Shemaila Sultana, James T. Inman, Smita S. Patel, Michelle D. Wang

We greatly appreciate the comments and suggestions from the reviewers to improve our manuscript. We have carefully considered each of the comments and have taken comprehensive action as detailed in the following response. We begin below with a summary of changes in the main text. This is followed by a detailed, point-by-point response to each comment. Each comment is shown in bold, and is followed by our response (not bold).

Summary of changes to the manuscript:

1. We have added data from a new control experiment, which shows that helicase alone is able to displace RNAP. The result indicates that helicase was the primary driver of the helicase/non-replicating DNAP complex in displacing RNAP for data shown in Figs. 3 and 4. This set of data is now added as a new Supplementary Fig. 7.
2. We have expanded the Discussion section to discuss possible mechanisms of the unwinding enhancement by a non-replicating DNAP.
3. We have explicitly defined “non-replicating DNAP” under the Introduction and clarified this term further under Results.
4. We have also made a few other changes to the manuscript to improve clarity and these changes are indicated in our response to comments from the reviewers.

Reviewer #1 (Remarks to the Author):

This manuscript by Sun, Patel, Wang and colleagues provides interesting single molecule data showing 1) that DNAP interacting with the helicase can increase the unwinding rate and processivity and that 2) this interaction can be used to facilitate replication restart by displacing a head on transcription complex and reengaging the mRNA transcript to continue replication. Both of these results are exciting a novel and should provide a basis for testing similar features in higher

organisms. The manuscript is well written and sound and most of my comments below are of a genuine curiosity. However, there are certain aspect that need to be clarified to strengthen this manuscript.

We thank the reviewer for his/her positive remarks and we have addressed individual comments below.

1) Intro, what is the predicted frequency of a replication-TEC head on collision? How often would this be expected say in T7 or bacteria. Provide a reference for significance.

In prokaryotes, replisomes move approximately 12-fold faster (~600 nt/s) than RNA polymerases (RNAPs) (50 nt/s). In addition, RNAP often pauses at regulatory sequences and stalls at sites of DNA damage. As replication and transcription proceed simultaneously on the same template DNA, the two must inevitably collide. In fact, there are many lines of evidence *in vitro* and *in vivo* that support the occurrence of both the co-directional and head-on collisions (Mirkin., *et al*, Mol Cell Biol, 2005; Pomerantz., *et al*, Cell cycle, 2010, Helmrich., *et al*, Nat Struct Mol Biol, 2013; McGlynn., *et al*, Mol Microbiol, 2012).

Taking bacteriophage T7 as an example, the T7 chromosome is a linear dsDNA, consisting of 39,936 bps. The growth cycle of T7 is remarkable: within 15 min of infection into an *E. coli* cell, the T7 DNA molecule is duplicated more than 100 times. Thus, it only takes a very short period of time to complete one round of replication. In addition, there are 17 T7 and *E. coli* RNAP promoters on the T7 genome that direct transcription. Moreover, it has been demonstrated both *in vitro* and *in vivo* that a T7 replisome uses an RNA transcript from T7 RNAP to initiate replication at the T7 replication origin (Hinkle, *et al.*, J Virol, 1980; Romano, *et al.*, PNAS, 1981). In light of these facts, one can envision that the collision between replication and transcription in bacteriophage T7 must happen quite frequently and T7 replisome has to effectively resolve these collisions, though we are unable to provide a concrete number here.

We greatly appreciate the reviewer's suggestion and have revised the introduction section to emphasize the significance of the collision on page 2 in the main text:

"In addition, as replication and transcription proceed simultaneously on the same template DNA, the two must inevitably collide. In fact, many lines of evidence *in vitro* and *in vivo* support the occurrence of both co-directional and head-on collisions."

2) "non-replicating DNAP" needs to be defined from the start. I believe you are using cat- mutant of gp5, but this should be explicitly stated. What happens when experiment in Fig 1c is performed with WT gp5 but in the absence of dNTPs. I imagine the result is the same? Also later you use "nonreplicating" when

describing an inverted dT. You should be specific in each case.

We apologize for not being explicit in defining this term. In this study, we define “non-replicating DNAP” as DNAP that is competent for replication but is in a state that is not synthesizing.

To clarify these, we now explicitly define “non-replicating T7 DNAP” on page 4:

“Non-replicating T7 DNAP is herein defined as DNAP in a state that is not replicating due to either lack of a complete set of dNTPs or lack of an extendable primer.”

In the experiments outlined in this manuscript, a “non-replicating DNAP” was ensured via two different approaches. For the experiments shown in Figs. 1 and 2, we only used one type of NTP (either ATP or dTTP) as the fuel source to power helicase unwinding, and DNAP replication is unable to proceed due to the lack of a full complement of dNTPs. For the experiments shown in Fig. 3, we modified the DNA primer so that DNAP cannot replicate prior to picking up an RNAP primer from the RNAP. The DNA primer modification contain an inverted dT to the 3' of the primer from which DNAP could not synthesize even in the presence of all dNTPs.

On page 5, we revised the manuscript to clarify why DNAP is not replicating for data shown in Fig. 1:

“We thus examined T7 helicase’s slippage behavior and unwinding activities (rate and processivity) in the presence of DNAP and 2 mM ATP. This nucleotide condition supports DNA unwinding but does not support DNA replication because dNTPs are missing.”

On page 9, we revised the manuscript to clarify why DNAP is not replicating prior to encountering the TEC for data shown in Fig. 3:

“In the first experiment, the leading strand was provided with a DNA primer containing an inverted dT at its 3' end that does not support DNA synthesis even in the presence of all dNTP.”

and

“To directly examine the consequences of collision between the helicase/non-replicating DNAP and a TEC, we used a parental DNA template that contained an inverted dT primer and a co-directional TEC stalled at +20 nt from its promoter (Fig. 3b). There should only be unwinding without DNA synthesis prior to their collision.”

3) This is sentence seems wrong? "Once helicase activity was detected, then helicase catalyzed unwinding was monitored under a constant force, which was not

**sufficient to mechanically unwind the fork junction without a helicase (Fig. 1b)."
without a DNAP?**

We apologize for not being clear here. We aimed to express that the constant force used in this assay was not sufficient to mechanically unzip the fork junction and the unwinding observed was a result of helicase activity instead of the force. To clarify that, we revised the text by stating:

“Once helicase activity was detected, then helicase catalyzed unwinding was monitored under a constant force, which was not sufficient to mechanically unzip the fork junction.”

4) What was the difference in rate at 8pN between Fig 1b and c? Is it the same as in Fig 3a?

We apologize for any confusion in Fig. 1. The experiments in Fig. 1b and 1c were conducted in 2 mM ATP at 8 pN and no replication occurred in both cases, whereas the experiments in Fig. 3a were performed at 0.5 mM dNTP (each) at 5 pN and the replication indeed occurred when DNAP is present. The unwinding rates at 8 pN in the presence and absence of non-replicating DNAP are shown in Fig. 1e. The helicase unwinding rates were not the same under these two experimental conditions.

The difference in the helicase unwinding rates is not surprising. As shown in our previous publications (Johnson., *et al*, Cell, 2007; Sun., *et al*, Nature, 2011; Sun., *et al.*, Nat Comm, 2015), helicase unwinding activity depends on NTP concentrations, the external force, and the presence of DNAP.

5) Page 7, Fig 2c, is the height of the force peak reproducible in those 75% of traces. Can't the magnitude be used to quantify the strength of the interaction of helicase-DNAP with DNA?

The peak force distribution is shown in Supplementary Fig. 5c. It has a mean force of 26 ± 1 pN (mean \pm s.d.). Some spread in the force is expected as the disruption process is thermally activated and is stochastic in nature.

The reviewer is correct to point out that this peak force may indeed be used to characterize the strength of helicase-DNAP interactions. The proper characterization, however, is quite non-trivial and requires “dynamic force spectroscopy (DFS)”. In order to map the interaction energy landscape using DFS, we need to repeat the disruption experiments under a broad range of stretch rates and establish proper theoretical modeling to describe the disruption process. We have previously used DFS to quantify histone-DNA and restriction enzyme-DNA interactions (Brower-Toland *et al.*, PNAS, 2002; Koch *et al.*, Phys. Rev. Lett., 2002). Although DFS studies are beyond the scope of the current work, it is potentially a powerful method to

study interactions in a replisome and we are interested in exploring this approach in the future.

6) Page 8, there does not seem to be proof for "DNAP-helicase interactions were transiently lost at the moment of detection". Couldn't it just as easily be that DNAP lost contact with DNA, but remained bound to helicase in those cases. The probability seems to low that 15% of the molecules were lost just as you were measuring force change?

We thank the reviewer for providing this explanation and we agree that this is a possibility. We have incorporated this explanation for our data in the text on page 8:

"Because we observed a complete lack of slippage during helicase unwinding in the presence of DNAP (Fig. 1), we favor the possibility that DNAP was always present at the fork, but the DNAP-helicase or DNAP-DNA interactions were transiently lost at the moment of detection."

7) Page 9, would be nice to see an example without any detecable pausing.

We thank the reviewer for this suggestion. A lack of pausing could be a result of an absence of a TEC and this could occur, for example, due to incomplete TEC formation. We feel that showing such a trace may confuse the readers.

8) Page, 9, any correlation of length of pause with increased rate of 2nd rate of synthesis. For example is a pause needed to get a change in rate correlating with TC-restart?

After reading the Reviewer's comment, we checked all of the traces and there does not appear to be any correlation between pause duration and subsequent re-initiation.

9) Page 10, Fig 4a, lanes with RNAP and DNA but no helicase. Looks like there is a small amount of runoff product here. But you say in text, "However, no run-off products were observed with dNTPs on a fork substrate when helicase was absent (Fig. 4a)."

The reviewer is correct to point out that a small amount of run-off product (~1.2%) was detected on the fork substrate in the absence of the helicase. We have added the following sentence on page 11:

" A very small amount of run-off product (~1.2% at the end of 10 minutes) was observed in the absence of helicase on forked TEC (Fig. 4a)"

10) Fig 4, would have been nice to have a lane with RNA primer only to more

clearly see the products you describe (i.e. TEC+dNTP +1 and +2 products). Also why is there 2 markers corresponding to 10mer? This is confusing. I realize that RNA and DNA can run differently, but we need to be clear what we are looking at?

We thank the reviewer for the comment. We show the RNA primer (fluorescein labeled) and dNMP incorporation by the TEC in Supplementary Fig. 9. The experiments in Fig. 4 were performed with non-radioactive primer and products are labeled with [α - 32 P]-dGTP. According to the template sequence, the first [α - 32 P]-dGMP is at 14-mer. Comparison of the reactions with fluorescent primer (Supplementary Fig. 9), indicates that the two bands observed between the 10-mer markers in Fig. 4 are 14-mer and 15-mer.

We used a 10-bp double stranded DNA ladder, and our denaturing sequencing gel is able to resolve the two strands. This is now indicated in the Methods.

11) Top of page 11, "we estimated that ~ 14% of the RNA had fully extended" Estimated or quantified?

We have corrected it to "quantified" in the manuscript (page 11). The method used for quantification is described in the Methods section.

12) WT T7 gp5 should also be included in methods.

Both wild-type and exo⁻ gp5 have now been included in the Methods section:

"Wild-type T7 gp5 and exo⁻ gp5 were purified as previously described (Patel *et al.*, JBC, 1992; Patel *et al.*, Biochemistry, 1991)."

Reviewer #2 (Remarks to the Author):

The manuscript entitled "Helicase promotes replication re-initiation from and RNA transcript " by Sun et.al. is an exciting and impactful paper by leading experts in the field, continuing a very strong collaboration between these two groups. The paper should draw interest from a wide variety of labs in diverse fields of replication, transcription and repair. The data are quite nice, and the important conclusions are supported by the data. I have a few comments below that might improve the information content for the readership, but overall the manuscript is quite acceptable in my opinion.

We thank the reviewer for his/her positive remarks and we have addressed individual comments below.

Specific comments:

1. It seems that the Intro of the paper could be a bit more informative about primase induced restart, especially considering the T7 helicase is the T7 primase and I don't think this was mentioned in the paper. One might presume the gp4 helicase-primase can prime both strands, as shown previously in the E coli and eukaryotic systems – but I'm not sure whether this has been demonstrated directly for T7 gp4. Maybe T7 gp4 simply can't do this? It this line of thought, it would be easy and useful to explain to readers the data/observations that eliminate T7 gp4 primase as a source of RNA priming in the experiments.

T7 primase requires single-stranded DNA for its priming activity, and there are no reports of it priming leading strand synthesis. The priming activity requires ATP+CTP, which were not added in our reactions.

We have also included the following sentence in the Results on Page 10:

“T7 gp4 also contains a primase domain, but this cannot be the source of primers in our experiments due to lack of priming nucleotides, ATP and CTP.”

In addition, we have also added a sentence to introduce the components of T7 replisome under Introduction on Page 3:

“It consists of T7 DNA polymerase (gp5 protein), T7 helicase-primase (gp4), processivity factor *E. coli* thioredoxin (trx), and the single strand binding protein (gene 2.5 protein). For simplicity, here the gp5/trx complex has been referred to as DNAP.”

2. The single molecule experiments use E. coli RNAP, but considering the genome is T7 and encodes its own RNAP, could the authors comment on whether T7 replisome will be expected to encounter E coli RNAP transcripts during T7 replication? Possibly the mRNA takeover does not require protein-specificity of interaction – and this reviewer actually finds that interesting and plausible. But however the authors would like to comment on it is fine, but seems noteworthy to point out.

We thank the Reviewer for pointing this out. During the early phase of T7 transcription, *E. coli* RNAP transcribes T7 genes through specific promoters (Kruger *et al.*, Microbiol Rev. 1981; Brautigam *et al.*, J Virol. 1973). Thus, there is a possibility of T7 DNAP to encounter *E. coli* RNAP. It also appears from our studies that the mRNA takeover does not require protein-specific interactions, because we observed replication re-initiation from both *E. coli* and T7 TECs.

We have commented on this under Discussion:

“During the early phase of T7 transcription, *E. coli* RNAP transcribes T7 genes through specific promoters. Thus, in addition to encounters of the T7 replisome with T7 RNAP, there is a possibility of encounters with the *E. coli* RNAP. Our study shows that both T7 and *E. coli* TECs can support T7 replication re-initiation, indicating that protein-specific interactions are not required.”

3. The ensemble work uses T7 RNAP and it would be useful to explain to readers the rationale for using the different RNAP's for single molecule vs ensemble experiments. I presume each RNAP has characteristics that make the use of one better for single molecule work and the other for ensemble studies?

We thank the reviewer for the suggestion. The primary reason that we chose to use *E. coli* RNAP in our single-molecule experiments is that TECs formed with T7 RNAP are not very stable (Mentesana *et al.*, JMB, 2000; Gopal *et al.*, JMB, 1999), making it unsuitable for our single-molecule experiments under force. In the last decade, our lab has fully characterized the kinetics and dynamics of the *E. coli* RNAP in details in single molecule condition and we found that paused *E. coli* TECs are extremely stable (Ma *et al.*, Science, 2013; Le *et al.*, Cell, 2018). In addition, this experimental design is also biologically relevant (please see our response to Reviewer #1's comment 1).

4. It would be informative to mention the “n” of experiments performed for the various experiments, perhaps just a text note in the figures themselves – but otherwise in the legends or main text.

We thank the reviewer for the suggestion. The # of traces for each experiment has now been stated either in the figure legends or in the text.

Reviewer #3 (Remarks to the Author):

“Helicase Promotes Replication Re-initiation from an RNA Transcript” by Sun et al. reports on new single-molecule optical tweezers experiments—supplemented by bulk biochemical assays—on T7 helicase and DNA polymerase (DNAP), essential components of the T7 replication machinery. In this work, the authors claim that a non-replicating T7 DNAP can enhance the activity of T7 helicase by reducing its slippage, leading to a faster and more processive unwinding. This enhancement is shown to be mediated through direct interactions between helicase and DNAP, with the complex formed across both strands of the DNA duplex at the DNA fork. Furthermore, the authors show that the helicase-DNAP complex can displace a stalled RNA polymerase (RNAP) and use a nascent RNA transcript as a primer to re-initiate DNA replication, at an efficiency of approximately 15%.

Overall, this is a very interesting manuscript that sheds new light on the synergy between replicative helicases and polymerases, and provides more evidence for

the broader role these two proteins may play. Generally the approach appears sound and several of the experiments are elegantly conceived. Also, bulk assays nicely complement the findings on replication re-initiation. The findings should be of broad interest to the scientific community working on replication and on molecular machines. Overall, the work is publishable, but there are several important points (outlined below) that need to be addressed before a final recommendation can be made.

We thank the reviewer for his/her positive remarks and we have addressed individual comments below.

Main questions:

1. Overall it was unclear how the authors ensured that DNAP was non-replicative in the measurements in Fig. 1, 2, and part of 3. Are dNTPs withheld in these measurements to block synthesis? If so this was not stated in the text. Or, is this claim made solely based on the lack of primer to initiate replication? If the latter, is there any effect of dNTPs on the observed DNAP-mediated enhancement in helicase activity? According to their model, the helicase is essentially pulling DNAP along the DNA; is DNAP then able to bind and incorporate dNTPs?

Upon reading the Reviewer's comment, we recognized that we did not clearly explain "non-replicating DNAP". We have made the following changes to the manuscript to clarify this.

(1) On page 4, we now explicitly define "non-replicating T7 DNAP":

"Non-replicating T7 DNAP is herein defined as DNAP in a state that is not replicating either due to lack of a complete set of dNTPs or lack of an extendable primer."

(2) On page 5, we revised the manuscript to clarify why DNAP is not replicating for data in Fig. 1:

"We thus examined T7 helicase's slippage behavior and unwinding activities (rate and processivity) in the presence of DNAP and 2 mM ATP. This nucleotide condition supports DNA unwinding but does not support DNA replication because dNTPs are missing."

(3) On page 9, we revised the manuscript to clarify why DNAP is not replicating prior to encountering the TEC for data shown in Fig. 3:

"In the first experiment, the leading strand was provided with a DNA primer containing an inverted dT at its 3' end that does not support DNA synthesis even in the presence of all dNTP."

and

“To directly examine the consequences of collision between the helicase/non-replicating DNAP and a TEC, we used a parental DNA template that contained an inverted dT primer and a co-directional TEC stalled at +20 nt from its promoter (Fig. 3b). There should only be unwinding without DNA synthesis prior to their collision.”

2. On a related point, what was the concentration used for each dNTP in the measurements? In the methods section, it is stated that the “concentrations [are] specified in the text”, but they are not.

We apologize for this typo. The dNTP concentrations were actually listed in the figure legends. We corrected this in the Methods section by stating:

“The unwinding and replication buffer consisted of 50 mM Tris-HCl pH 7.5, 40 mM NaCl, 10% glycerol, 1.5 mM EDTA, 2 mM DTT and dNTPs at the concentrations specified in the figure legends, and MgCl₂ at a concentration of 2.5 mM in excess of the total nucleotide concentration.”

3. In general there is a lack of discussion on a potential molecular mechanism of enhancement. Could the authors speculate how the non-replicative DNAP acts as a barrier against helicase slippage? What prevents DNAP itself from slipping backward when it is not elongating a DNA strand, i.e. when there is no physical barrier preventing it from moving backward?

We thank the reviewer for the suggestion. Our studies show that the stimulation of the unwinding activity of T7 helicase by non-replicating DNAP is mediated by direct interactions between them. A unique feature possessed by T7 replisome is that T7 helicase and DNAP interact physically through several modes (Kulczyk, *et al.*, 2012; Zhang, *et al.*, 2011). One of the most important interaction modes between these two proteins is an electrostatic interaction via the acidic C-terminal tail of helicase and basic thioredoxin-binding domain of DNAP. It increases the processivity of the replisome from 5 to 17 kbp by capturing DNAP that transiently dissociates (Hamdan, *et al.*, 2007). In addition, we previously demonstrated that this interaction also ensures that a T7 replisome directly synthesizes through a DNA lesion (Sun, *et al.*, 2015). In light of these results, we speculate that both non-replicating DNAP and helicase bind on DNA separately during helicase unwinding and interact with each other via specific domains. When helicase loses grip on DNA, the DNA-bound non-replicating DNAP may retain helicase on the DNA template via DNAP-helicase interactions, thus preventing helicase slippage or dissociation. Furthermore, the binding of the non-replicating DNAP at the fork may also help relieve the regression pressure from the fork and increase helicase unwinding rates.

Following the reviewer's suggestion, we have incorporated this under the Discussion section.

4. In the text, the authors state that they “anticipate that in the absence of force, helicase’s unwinding activities may also be stimulated by the presence of a non-replicating DNAP.” As shown in Fig. 1e, the effect of force on unwinding rate is significant, with no unwinding detected below 4 pN (presumably, the signal is too low to detect below this force). Although the authors show zero-force, bulk measurements of DNA synthesis, no bulk unwinding assays are shown demonstrating DNAP enhancement at zero force. The authors should address this.

The single molecule results on stimulation of helicase unwinding rates by non-replicating DNAP are supported by our previous bulk experiments (Nandakumar, et al., *Elife*. 2015). In the absence of dNTPs (with dTTP only as fuel for helicase activity), the catalytic efficiency (k_{cat}/K_m) of the helicase was increased by ~44% in the presence of the DNAP. The following sentence has been included on Page 6:

“This is also supported by our previous bulk study where T7 DNAP enhanced the catalytic efficiency of unwinding by T7 helicase in the absence of dNTPs (with dTTP only as fuel for helicase activity).”

5. In Fig. 1e, the authors claim that DNAP increases the unwinding rate of T7 helicase for forces <9 pN but has no effect for forces >9 pN. The authors should make clear how statistically significant the difference in rates (with and without DNAP) is below 9 pN, as this statement was not particularly convincing. Are the error bars shown in the figure standard error of the mean or standard deviation? This should be made clear in the figure caption, and a statistical analysis provided to assess the significance of any observed differences in rates.

We thank the reviewer for pointing this out. The error bars shown in Fig. 1e represent standard deviations. We have now stated so in the figure legend. We use standard deviation instead of standard error of the mean as error bars in order to give a conservative estimate of the errors in the measurement. These errors could be a result of systematic errors that cannot be reduced by averaging.

We agree with the Reviewer that a definitive statement on the lack of differences in the rates above 9 pN requires a statistical argument. We have therefore revised our statement on page 6:

“At forces below 9 pN, helicase alone unwound with frequent slippage, and DNAP increased the helicase unwinding rates between slips, whereas this effect was not detectable within measurement uncertainties in a higher force region (Fig. 1e): student t-test $t(7) = -0.65$ (10 pN) and $t(6) = 0.87$ (12 pN), $p > 0.05$.”

Minor points:

6. In the section “Helicase assists a non-replicating DNAP in displacing RNAP and re-initiating replication”, second paragraph, the authors state that “the extension was monitored under a low constant unzipping force”. The force or force range should be stated explicitly.

We apologize for not making this clear. Again, the force was shown in the figure legends. We took the reviewer’s advice and rephrased the sentence as follow on Page 9:

“we carried out two control experiments to characterize the rates of extension under a constant unzipping force of 5 pN.”

7. In the section “Non-replicating DNAP with a helicase localizes to the fork junction” the authors describe experiments in which the DNAP-helicase complexes are ruptured by force (Fig. 2). For complexes that were presumably broken by applying force, did the authors observe resumption of unwinding after the force was lowered again? If so, was slippage observed, pointing to the helicase no longer making physical contact with DNAP?

We thank the reviewer for the suggestion. To test whether the DNAP still interacts with helicase after the disruption, we performed the following experiment: after the initial disruption, we lowered the force and monitored helicase unwinding. We then performed a second disruption. We observed that a fraction of traces showed a disruption peak in the second disruption. However, this fraction (20 %) was much smaller than the fraction at the initial disruption (75%). These data suggest that the force might disrupt DNA-DNAP or DNAP-helicase interactions, resulting in the absence of the second peak.

8. In the measurements of RNAP displacement by the DNAP-helicase complex, is it clear whether DNAP or helicase are displacing RNAP? Is it known whether T7 helicase alone or replicating DNAP alone can displace RNAP or is the synergy of the DNAP-helicase complex required?

We performed a control experiment to test which protein plays a major role in displacing RNAP. Under our experimental condition (5 pN and 0.5 mM dNTP), we found that even when DNAP is provided with a DNA primer that can be extended, DNAP alone is unable to efficiently replicate as the fork presents as a major barrier for DNAP under this low unzipping force (Sun *et al.*, Nat Commun, 2015). In contrast, under the same condition, T7 helicase alone is able to unwind and displace RNAP. Thus, we conclude that T7 helicase is primarily responsible for displacing RNAP and this displacement was further facilitated by synergistic interactions of helicase with a non-replicating DNAP.

These data are now included as a new Supplementary Fig. 7 and are discussed on page 10.

Reviewers' Comments:

Reviewer #1:

Remarks to the Author:

The authors have responded all of my concerns. I would recommend publication after minor comments.

One final suggestion.

10) Indicating size of intermediate products (14, 15mers) on the left side of the gel may be helpful. Also, I was really asking about the 10mer label in the markers in lane 1 (Why 2 bands?)

And: with the addition of Sup Fig 7. Why is there no slippage occurring with helicase alone at 5pN but there is at 8pN (Fig 1b)? Is it that the higher force displaces the excluded strand more allowing slippage, while at lower force, the helicase interacts with both strands preventing slippage? I didn't see helicase only experiments with lower forces in ref19.

Reviewer #2:

Remarks to the Author:

I am completely satisfied by the revised manuscript and the responses to the reviewers. The revised manuscript is now suitable for publication.

Reviewer #3:

Remarks to the Author:

In their revision to "Helicase Promotes Replication Re-initiation from an RNA Transcript" Sun et al. have addressed all the reviewer comments satisfactorily.

This is an interesting manuscript that should of interest to the scientific community working on the replication machinery and beyond. The work is sound and recommended for publication in Nature Communications.

Response to Reviewers

We would like to thank the reviewers for their additional comments. We have carefully considered the remaining comments of Reviewer #1 and have a detailed, point-by-point response to each comment below. Each comment is shown in bold, and is followed by our response (not bold).

Reviewer #1 (Remarks to the Author):

The authors have responded all of my concerns. I would recommend publication after minor comments.

We thank the reviewer for his/her positive remarks and we have addressed individual comments below.

One final suggestion.

10) Indicating size of intermediate products (14, 15mers) on the left side of the gel may be helpful. Also, I was really asking about the 10mer label in the markers in lane 1 (Why 2 bands?)

We thank the reviewer for the suggestion. We have indicated the intermediate products on the side of the gel now.

We observed a double band for the 10mer in the DNA ladder lane because our denaturing sequencing gel is able to resolve the two strands of the 10mer into separate bands, likely due to the slight sequence difference between the two strands. This is now indicated in the Methods.

And: with the addition of Sup Fig 7. Why is there no slippage occurring with helicase alone at 5pN but there is at 8pN (Fig 1b)? Is it that the higher force displaces the excluded strand more allowing slippage, while at lower force, the helicase interacts with both strands preventing slippage? I didn't see helicase only experiments with lower forces in ref19.

Previously we found that T7 helicase unwinds but slips in the presence of rNTPs; whereas dNTPs support processive unwinding without slippage (Sun et al., *Nature*,

2011). In Fig. 1b, we observed slippage behavior of T7 helicase because only ATP was present. In the force range from 5 pN to 8 pN, T7 helicase always slips when it is only powered by ATP (data not shown). In Supplementary Fig. 7, we performed the helicase unwinding experiment in the presence of all four dNTPs which support processive unwinding of T7 helicase without slippage.

Reviewer #2 (Remarks to the Author):

I am completely satisfied by the revised manuscript and the responses to the reviewers. The revised manuscript is now suitable for publication.

We thank the reviewer for his/her positive remarks.

Reviewer #3 (Remarks to the Author):

In their revision to "Helicase Promotes Replication Re-initiation from an RNA Transcript" Sun et al. have addressed all the reviewer comments satisfactorily.

This is an interesting manuscript that should of interest to the scientific community working on the replication machinery and beyond. The work is sound and recommended for publication in Nature Communications.

We thank the reviewer for his/her positive remarks.